# ROYAL SOCIETY
# OPEN SCIENCE

electrical engineering

wireless communications, orbital angular momentum multiplexing, antenna arrays, spectral efficiency

**Author for correspondence:**
Ben Allen
e-mail: ben.allen@eng.ox.ac.uk

# Experimental evaluation of spectral efficiency from a circular array antenna producing a Laguerre–Gauss mode

Ben Allen[1,2], Timothy D. Drysdale[3] and Chris Stevens[1]

[1]Department of Engineering Science, University of Oxford, Parks Road, Oxford OX1 3PJ, UK
[2]Network Rail, Elder Gate, Milton Keynes MK9 1ER, UK
[3]Institute for Digital Communications, School of Engineering, University of Edinburgh, Edinburgh EH9 3FG, UK

BA, 0000-0002-6308-8383

We present the four-dimensional volumetric electromagnetic field measurements ($x$, $y$, $z$ and frequency) of the complex radiated field produced by an 8-element circular antenna array. The array is designed to produce a Laguerre–Gauss (LG) mode $l = +1$ over the frequency range of 9–10 GHz. We evaluate our findings in terms of far-field LG mode purity and spectral efficiency in terms of the quadrature amplitude modulation (QAM) modulation scheme that can be supported. The application of LG modes in radio systems is as a means of multiplexing several data streams onto the same frequency, polarization and time slot, thus making a highly spectrally efficient transmission system or enhancing radar systems by means of exploiting mode behaviour as an additional degree of freedom. Our results show that for the circular antenna array, we find that mode purity is sufficient to support binary phase shift keying or quadrature phase shift keying modulation over a 0.3 GHz bandwidth, which corresponds to a spectral efficiency of 1.5 b s$^{-1}$ Hz$^{-1}$ per mode. Closer to the antennas' design frequency, 256QAM modulation may be supported over a 0.05 GHz band, and which corresponds to a spectral efficiency of 11 b s$^{-1}$ Hz$^{-1}$ per mode. We anticipate the practical insights provided in this paper contribute to the successful design of such systems.

## 1. Introduction

Orbital angular momentum (OAM) radio has caught the eye of researchers since 2012, when Tamburini *et al.* [1] described an

**Figure 1.** Amplitude and phase signatures of LG modes $l = 0$, $-1$ and $+2$.

experiment that demonstrated how OAM modes may be used to multiplex two radio signals on the same polarization, frequency and time. Prior to that, OAM had been investigated for optical tweezers [2], optical free space [3] and fibre optic communications links [4]. More recently, applications of OAM have been investigated as a means of enhancing radar systems [5]. For both communications and radar systems, the signals comprise one or more independent modes with particular spatial distributions of the phase [6]. This family of modes are termed Laguerre–Gauss (LG) modes, where their normalized complex amplitude can be expressed in cylindrical coordinates ($r$, $\phi$, $z$) around the axis of propagation as follows [7]:

$$\text{LG}_{p,l}(r,\phi,z) = \frac{1}{w(z)} \sqrt{\frac{2\rho!}{\pi(|l| + \rho)!}} \left(\frac{\sqrt{2}r}{w(z)}\right)^{|l|} L_p^{|l|}\left(\frac{2r^2}{w^2(z)}\right)$$
$$\times \, e^{-(r^2r^2/w^2(z))} \times e^{-j((kr^2/2R_{c(z)})-l\emptyset-(2p+|l|)\varphi(z))},$$

(1.1)

where $p$ and $l$ are the radial and azimuthal indices and $l$ is often referred to as the OAM mode number, $k$ is the wavenumber, $R_c(z)$ is the curvature of the phase-front at position $z$ along the axis of propagation, $\phi = \arctan(z\lambda/\pi w_o^2)$ is the Gouy phase, $w(z)$ is the beam radius and $w_o = w(z = 0)$ is the beam waist. Finally, $L_p^{|l|}$ is a Laguerre polynomial of order $l$ and degree $p$.

An example of the two-dimensional fields created by modes $l = -0$, $-1$ and $+2$ are shown in figure 1 when observed along the axis of propagation. For mode $l = 0$, the phase and amplitude signatures are uniform across the plane of interest, whereas for mode $l = -1$, the phase signature exhibits an increment of $2\pi$ radians around a circle with centre and the middle of the figure and the amplitude pattern exhibits a 'doughnut' shape. For mode $l = +2$, the phase signature exhibits a rotation of $4\pi$ radians in the other direction and the amplitude signature is also a 'doughnut' shape but of wider diameter. There are, theoretically, an infinite number of modes following the described trend, but only a small number have been realized owing to practical constraints such as array size, feed structures and realizing required beamwidths. A detailed description of LG modes is given in [8].

Because these modes are mutually independent, they may be used for multiplexing several data streams that occupy the same frequency, polarization and time slot. For this reason, they have attracted the attention of several researchers such as [9–11], where effort has been put into realizing and characterizing experimental radio links based on OAM multiplexing. Because independence only holds when observed multiple points along the axis of propagation, OAM mode multiplexing is most suited to point-to-point communication links such as for data back-hauling. For example, a team from Nippon Telegraph and Telephone used 10 LG modes to obtain a net data rate of 100 Gb s$^{-1}$ over a point-to-point radio link operating at 28 GHz [12]. A number of works have reported on the measured field signatures observed at a particular point along the axis of propagation and have used this to determine the mode purity exhibited in practice [9,11,12]. The modes may be visualized as a mode 'spectrum' and enables the mode purity to be determined. This is important because, in practice, energy can spread from the wanted mode into other modes because of: mutual coupling; multi-path;

finite antenna geometry, miss-alignment, etc. This causes crosstalk between the data streams and hence limits performance. So far it has proved very challenging to achieve the levels of mode purity required to support high data rates without using interference cancellation techniques, as is the case described in [13]. Careful antenna and mode multiplexer design and realization is required, together with careful link alignment, and a goal of achieving a signal-to-noise plus interference ratio (SNIR) of at least 30 dB is highly desirable for supporting higher modulation orders.

Hamilton *et al.* [14] presents an analysis of three-dimensional fields produced by an OAM array antenna operating in the 4–6 GHz band. The focus of that work was on the effect of beam collimation by means of a lens. By contrast, this paper examines the electromagnetic far-field (FF) signature of an LG mode exhibited in the 9–10 GHz band in terms of both mode purity and spectral efficiency where the FFs are determined by means of a transform performed on the near field (NF) measurements. We present results from measurements taken of the electromagnetic field emitted from a circular antenna array configured to produce an OAM mode $l = +1$ at a nominal frequency of 9.6 GHz. The measurements have enabled the FF to be determined over a 1 GHz bandwidth. Furthermore, the measurements have allowed a 'cut' across the axis of propagation to be shown in the NF. The complex NF is transformed to provide the FF complex patterns. Performance is determined in terms of mode purity and this figure of merit is then used to determine the Shannon capacity of the channel.

The sections of the remainder of this paper are as follows. Section 2 described the measurements system and methodology. Results of the measured field and mode purity are shown in §3. Conclusions are provided in §4.

## 2. Measurement system and methodology

The field measurements have been conducted over a volume with axis defined in figure 2. The $z$-axis is defined as the direction of propagation of the emitted field, where the antenna under test (AUT) is located at $x = y = z = 0$, which is located at the centre of the bottom plane of the volume, as indicated in figure 2.

The measurement system consists of a half-wave dipole probe antenna that has an $|S_{11}| > 10$ dB across the band of interest and mounted so that it was aligned with the primary polarization of the AUT. The probe was connected to one port of a vector network analyser (VNA) and the AUT was connected to the other. There are three stepper motors for controlling the $x, y, z$ location of the probe and these are connected to a PC that runs the scan sequence using a Visual Basic script. The probe was moved around the defined volume in 6.8 mm steps and cover an area of $14 \times 14 \times 7$ cm, amounting to 10 976 spatial measurements points. The maximum height of the measurements volume was 7 cm, or 2.5 $\lambda$ above the AUT. These measurements all occurred in the NF of the AUT and so the NF data were then transformed to the FF by means of a numerical transform described later in this section. For this work, we do not require the slight improvement in accuracy that would occur from implementing probe correction and leave that to future work. The VNA was configured to span 9–10 GHz in 267 frequency steps of approximately 3.75 MHz and measures the complex $S_{21}$ at each location. The VNA was calibrated with short, open, load and transmission references prior to measurements commencing. The measurement for each spatial location is stored in a file for analysis. The probe and measurement system are depicted in figure 3.

The AUT was an 8-element circular antenna array, as shown in figure 4 and full details of the dimensions are given in [15]. The array was designed to produce an LG mode $l = +1$ at a nominal frequency of 9.6 GHz and consists of eight rectangular patch antennas uniformly distributed around a circle. The incremental phase shift between elements required to produce the desired LG mode is realized by means of a 'corporate feed' network with the electrical length of each feed selected to provide the required phase shift and impedance to each element. The array is probe-fed from the rear by means of an SubMiniature version A socket.

Data analysis was conducted as follows. First, the 10 976 NF measurement files that cover the measurement volume and frequency range of interest were read. Data were then extracted according to the objective of the analysis. For example, a selected plane analysed as a function of frequency; or the fixed frequency and behaviour as a function of the spatial plane determined. The selected data were then split into amplitude and phase components and each analysed separately as a function of either location or frequency.

The FF data were generated by numerically propagating the NF data with the well-known scalar Green's function (such as described in Eqn 10.16 of [16]) to give the fields at a rectangular grid of 200

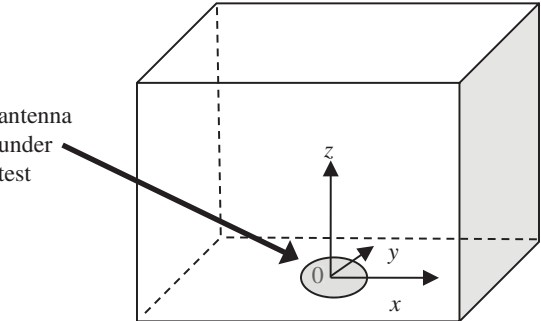

**Figure 2.** Definition of axis.

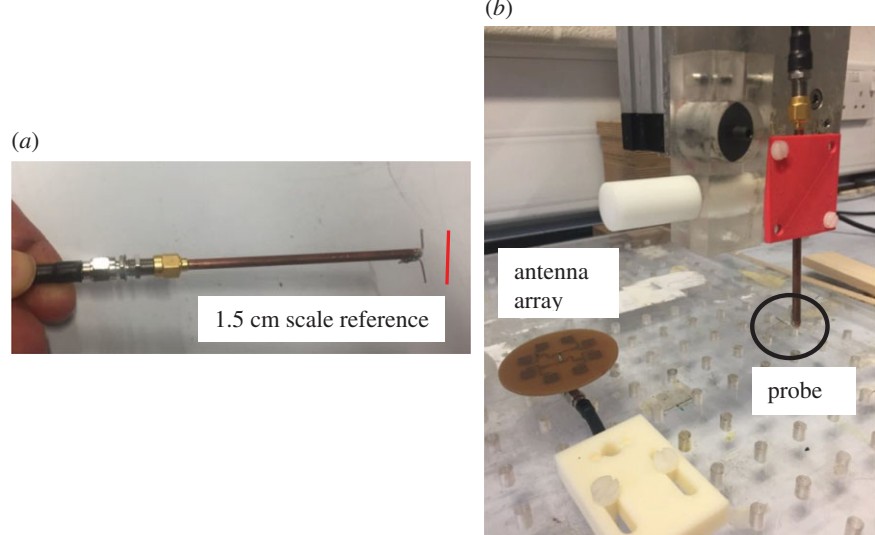

**Figure 3.** Measurement system. (*a*) probe (*b*) measurement volume.

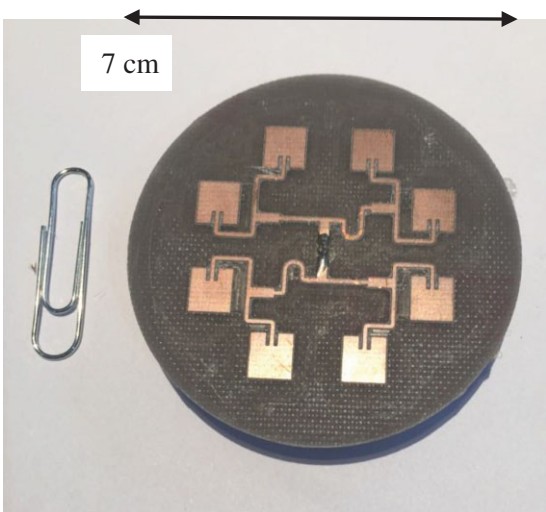

**Figure 4.** Antenna array (paper clip for size reference).

by 200 points lying on a two-dimensional planar surface 5 m away along the boresight, of overall lateral dimension 2.5 m by 2.5 m. The two-dimensional NF data are sufficient owing to the directional nature of the antenna, and for our scanner, it is impossible to measure fields on all sides of an enclosing volume in

any case. The complex FF data at each point $r$ is calculated by taking a sum of the contribution from all the sources in the near field at positions $r'_{mn}$ as follows [17]:

$$E_r(m, n) = \sum_{m=0}^{M-1} \sum_{n=0}^{N-1} \frac{e^{jk|r-r'_{mn}|}}{4\pi|r - r'_{mn}|}, \tag{2.1}$$

where $m$ is the $m$th $x$-position index, $n$ is the $n$th $y$-position index and $k = 2 \cdot \pi / \lambda$ is the wavenumber. The calculation is repeated separately for each frequency.

The FF mode spectra are calculated according to Eqn 9 in [10], where each mode spectrum comprises a set of coefficients representing the magnitude of each of the possible underlying pure modes. In this way, an antenna transmitting an ideal OAM mode has only a single non-zero coefficient at the intended mode number, whereas practical antennas usually show unwanted energy is transmitted on other mode numbers, thus limiting the overall signal to interference ratio (SIR). To aid comparison with other antenna designs, we do not consider other potential noise sources such as thermal noise associated with the receiving electronics. Variations of the mode purity profile are observed as the location of the $z$-plane changes.

# 3. Results

## 3.1. *X–Y* plane

The complex NF has been measured in the $X–Y$ plane for $Z_{min} \leq Z \leq Z_{max}$. The NF data at $Z_{min}$ has been transformed to the FF using the method described above. Figure 5 shows snap-shots of the resulting FF amplitude and phase patterns at 9.6 GHz, which is the centre frequency of the AUT defined as the frequency where the lowest return loss occurs. Figure 5$a$ shows the normalized amplitude pattern in dB, and figure 5$b$ shows the corresponding phase pattern. The circles marked on figure 5$b$ are used to extract the corresponding phase trajectories for each radius to enable comparison, shown in figure 5$c$ for each radius.

Figure 5$c$ shows the phase profile obtained from around each of the circles as well as the ideal profile for $l = +1$. Each profile has been normalized to start at 0 radians to allow for easy comparison. The figure shows more variations away from the ideal as the radius increases. We propose that these phase perturbations away from the ideal are caused by radiation and mutual coupling from the array feed network.

Figure 5$d$ shows the resulting mode number for each of the phase profiles, as well as the ideal $l = +1$ profile. Larger deviations are observed for larger radius phase profiles. The boundaries marking the region of the $l = +1$ mode are marked with adjacent modes being shaded, hence areas of the curves that show the mode profile straying into adjacent modes are seen, These curves have been obtained by using the method described in [9], where the mode is determined between two points on the given curve separated by angle $\beta 12$ using the following equation:

$$l = \frac{\theta_1 - \theta_2}{\beta_{12}}, \tag{3.1}$$

where $l$ is the mode computed between angle $\beta_{12}$ with phases $\theta_1$ and $\theta_2$ obtained at either end of angle $\beta 12$. The analysis increments from $\beta_{12} = 0$ through to $2\pi$ radians and the mode plotted for each increment.

## 3.2. *X–Y* plane versus frequency

The above methodology has been repeated for the $X–Y$-plane 'cut' with $z = Z_{max} = 7$ cm and the FF mode spectrum determined for each frequency step between 9 and 10 GHz.

The resulting mode spectrum is shown in figure 6. The strongest mode is clearly seen at $l = +1$ and holds across the 1 GHz frequency band, with some variations being present. The mode boundaries are marked on the figure by white lines which shows the boundaries with the two adjacent modes.

## 3.3. Signal-to-interference ratio

The SIR ratio is plotted in figure 7. The SIR is calculated from the point of view of an ideal OAM antenna receiving a signal from the AUT (i.e. having mode spectra as plotted in figure 6). The SIR is taken as the ratio of the power in mode $l = +1$, to the power in the rest of the modes (which are unwanted). This has been determined for each frequency step across the band, with the resulting SIR plot shown in figure 7. This shows the maximum SIR of 33.5 dB occurs at 9.65 GHz and a minimum SIR of 2.5 dB at 9.2 GHz.

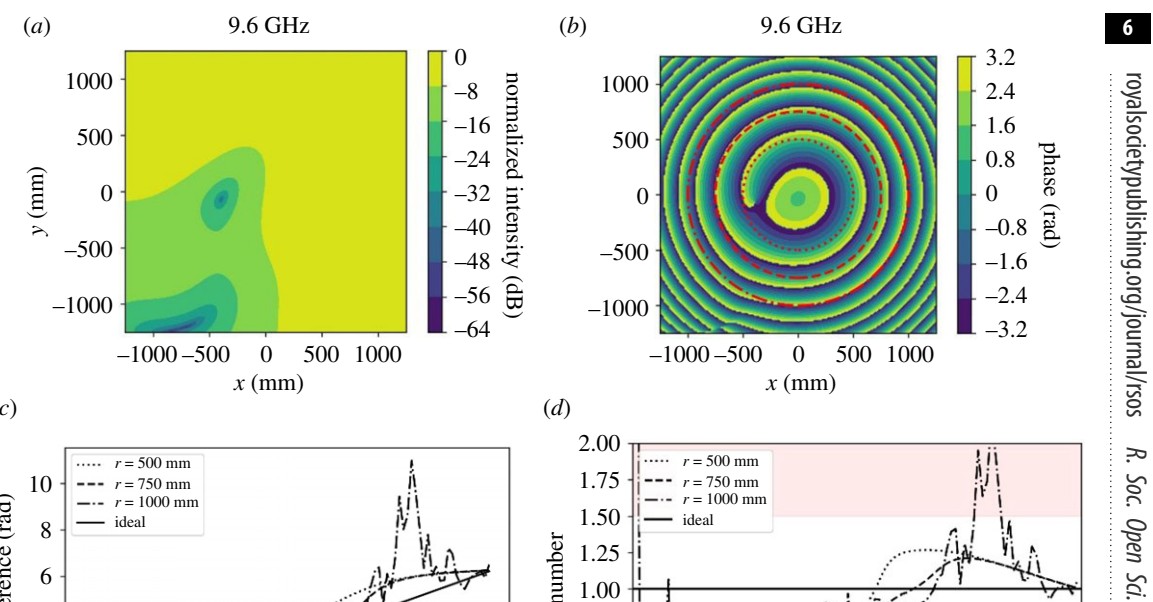

**Figure 5.** FF patterns and mode spectra at 9.6 GHz. (*a*) Amplitude signature (dB); (*b*) phase signature (radians); (*c*) phase gradient; and (*d*) mode versus azimuth angle.

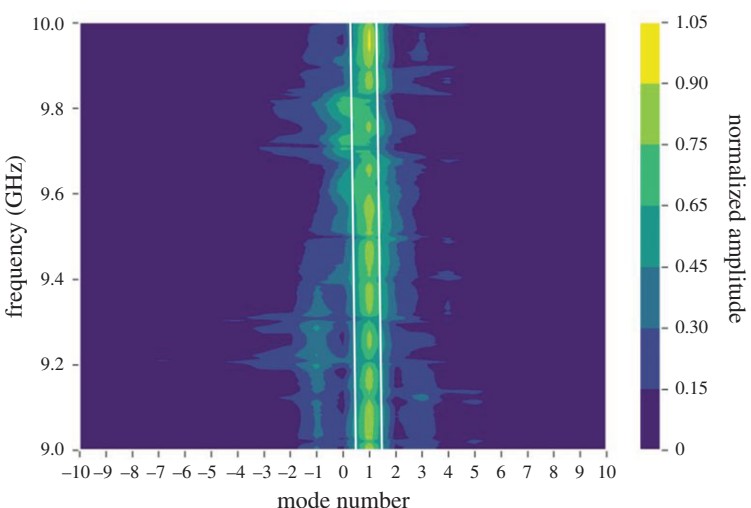

**Figure 6.** Mode spectrum versus frequency.

The resulting channel capacity can be computed using the Hartley–Shannon channel capacity formula [18] given by equation (3.2) where $\xi$ is the spectral efficiency in b s$^{-1}$ Hz$^{-1}$ and $S/I$ is the SIR. This assumes a memoryless channel; the noise is negligible compared to the interference power and the interference power distribution is Gaussian [19]. Because the AUT can support a single OAM mode and OAM multiplexing exploits a number of modes, we express spectral efficiency in terms of b s$^{-1}$ Hz$^{-1}$ per mode. Detailed analysis of this is beyond the scope of this work. We obtain spectral efficiencies between 1.5 b s$^{-1}$ Hz$^{-1}$ per mode and 11 b s$^{-1}$ Hz$^{-1}$ per mode over the band of interest:

$$\xi = \log_2\left(1 + \frac{S}{I}\right). \tag{3.2}$$

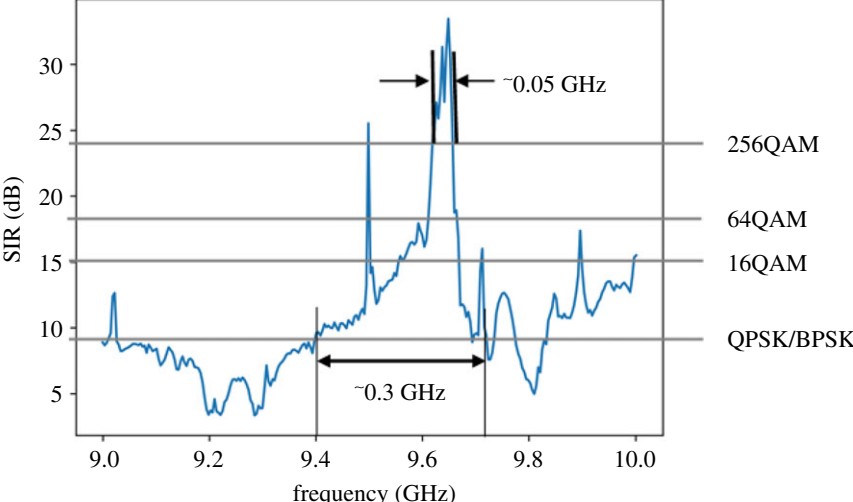

**Figure 7.** SIR and QAM modulation schemes for far-field data.

Assuming a maximum bit error rate of $10^{-5}$, this results in 256 quadrature amplitude modulation (QAM) modulation being supported where the SIR is maximum, through to binary phase shift keying (BPSK) where is SIR is at least 9 dB. Below this value, spectrally  efficient modulation cannot be supported, but power efficient-modulation schemes such as frequency shift keying can be [20]. The minimum SIR requirement for each modulation scheme are marked on figure 7, which also shows that 256QAM is achievable over a 0.05 GHz bandwidth and a modulation scheme of at least BPSK is achievable over a 0.3 GHz bandwidth. Given that the array is a narrowband design, it shows that OAM operation can be supported over a relatively wideband.

By way of comparison, we can consider a point-to-point link consisting of a single patch antenna at either end. We can expect a similar signal to noise ratio profile as a function of frequency, showing a peak over a narrow band corresponding to the design frequency of the patches. However, such a set-up would only be able to support a single mode ($l = 0$). By contrast, a system comprising multi-mode OAM antennas would be able to support $N$ modes and hence $N$ parallel data streams all on the same frequency. Hence this system would be $N$-times more spectrally efficient.

# 4. Conclusion

We show that modal analysis of four-dimensional field data allows a rapid estimate of SIR in LG antenna modes, leading to understanding which modulation schemes can be employed over what bandwidth. For the AUT, we obtain SIR > 9 dB over 0.3 GHz bandwidth, permitting BSPK/quadrature phase shift keying modulation to be used, corresponding to a total spectral efficiency of 1.5 b s$^{-1}$ Hz$^{-1}$ per mode. Closer to the antennas' design frequency of 9.6 GHz, SIR exceeds 30 dB over a 0.05 GHz band, although the antenna is narrowband, producing a spectral efficiency of 11 b s$^{-1}$ Hz$^{-1}$ per mode and allowing 256QAM modulation. This clearly demonstrates the value in our analysis for evaluating the performance of the AUT for supporting OAM radio links. Future work on antenna designs will continue to concentrate on obtaining a broadband SIR > 30 dB; our paper assists those researchers by providing a method to quantify the OAM performance of their antenna.

Data accessibility. Measured complex radiation pattern data used to produce the results in §3 are available from Dryad Digital Repository: https://doi.org/10.5061/dryad.44j0zpccc [21].

Authors' contributions. B.A. performed the measurements, drafted the manuscript, guided the work and performed some of the analysis. T.D.D. performed some of the analysis and has critically reviewed the manuscript. C.S. performed the measurements, has made substantial contributions to the concept and has critically reviewed the manuscript. All authors gave final approval for publication.

Competing interests. The authors declare no competing financial interests.

Funding. Ben Allen is grateful for the support of the Royal Society Industrial Fellowship scheme under award number (grant no. IF160001).

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
