## [Reviewer comments · Royal Society Open Science]

Review History

RSOS-201711.R0 (Original submission)

Review form: Reviewer 1

Is the manuscript scientifically sound in its present form?

Yes

Are the interpretations and conclusions justified by the results?

Yes

Is the language acceptable?

Yes

Do you have any ethical concerns with this paper?

No

Have you any concerns about statistical analyses in this paper?

No

Recommendation?

Accept with minor revision (please list in comments)

Comments to the Author(s)

In this paper authors presented an interesting concept of 4D volumetric electromagnetic field measurements. Results are validated by measurements. Some minor comments are:

- Add some quantitative results in the abstract.
- A comparison with state of the art should be added.
- Dimensions should be added in the figure 4 for each element.
- reference for equ 2 is missing
- legends in fig 5 are not readable

Review form: Reviewer 2

Is the manuscript scientifically sound in its present form?

Yes

Are the interpretations and conclusions justified by the results?

Yes

Is the language acceptable?

Yes

Do you have any ethical concerns with this paper?

No

Have you any concerns about statistical analyses in this paper?

No

Recommendation?

Accept with minor revision (please list in comments)

Comments to the Author(s)

The paper presents an experimental communication radio link spectral efficiency performance for a circular patch antenna array radiating the Laguerre-Gauss mode ($l=1$) over the 9-10 GHz. The results make a demonstrative and significant contribution to the application and performance of BPSK, QPSK and QAM modulation schemes using OAM (orbital angular momentum).

However, there are a number of minor typographical and presentation issues that need attention and clarification as follows;

- (i) p3, L55, need a full stop after "patterns".
- (ii) What is the impact on the radiation/polarisation pattern of the LG ($l=-1$) mode due to the corporate feed network radiation.
- (iii) Equation 2 the math function term is missing please insert.
- (iv) p7, L14, L20 need to be consistent in referring to figures either with a capital Fig or lower case fig.
- (v) p7, p.30 need to add equation number.
- (vi) Figure 6 right hand side scale label.
- (vii) p7,L27 full stop at "seen".

(viii) p8, L29 I can clearly see $l=+1$ mode but the $l=-1$ mode is not strongly excited and also doesn't appear to exist for all frequencies. Explain why?

(ix) The abstract states that you wish to excite and use the LG $l=-1$ mode however the $l=+1$ seems to dominate the mode spectrum in Figure 6 and forms the basis for the subsequent results in Fig. 7. Please explain?

(ix) p10,L17 The authors should comment or compare the demonstrated spectral efficiencies per mode in context with channel capacity from an equivalent patch antenna array radio link.

Decision letter (RSOS-201711.R0)

Dear Dr Allen

On behalf of the Editors, we are pleased to inform you that your Manuscript RSOS-201711 "Experimental Evaluation of Spectral Efficiency from a Circular Array Antenna producing a Laguerre-Gauss Mode" has been accepted for publication in Royal Society Open Science subject to minor revision in accordance with the referees' reports. Please find the referees' comments along with any feedback from the Editors below my signature.

Please submit your revised manuscript and required files (see below) no later than 7 days from today's (ie 16-Nov-2020) date. Note: the ScholarOne system will 'lock' if submission of the revision is attempted 7 or more days after the deadline. If you do not think you will be able to meet this deadline please contact the editorial office immediately.

on behalf of Dr Chong Li (Associate Editor) and R. Kerry Rowe (Subject Editor)

Associate Editor Comments to Author (Dr Chong Li):

Associate Editor: 1

Comments to the Author:

Dear Authors,

The paper demonstrates experimental evaluation of spectral efficiency for a circular antenna array with LG mode of radiation. This is interesting work and could potentially benefit new developments of antennas of this kind and therefore contribute to the development of wireless communications. This work should be published. However the reviewers have some comments especially Reviewer 2's questions ii, viii-ix that should be addressed carefully by the authors.

Associate Editor

Dr. Chong Li

Reviewer comments to Author:

Reviewer: 1

Comments to the Author(s)

In this paper authors presented an interesting concept of 4D volumetric electromagnetic field measurements. Results are validated by measurements. Some minor comments are:

- Add some quantitative results in the abstract.
- A comparison with state of the art should be added.
- Dimensions should be added in the figure 4 for each element.
- reference for equ 2 is missing
- legends in fig 5 are not readable

Reviewer: 2

Comments to the Author(s)

The paper presents an experimental communication radio link spectral efficiency performance for a circular patch antenna array radiating the Laguerre-Gauss mode ($l=1$) over the 9-10 GHz. The results make a demonstrative and significant contribution to the application and performance of BPSK, QPSK and QAM modulation schemes using OAM (orbital angular momentum).

However, there are a number of minor typographical and presentation issues that need attention and clarification as follows;

- (i) p3, L55, need a full stop after "patterns".
- (ii) What is the impact on the radiation/polarisation pattern of the LG ($l=-1$) mode due to the corporate feed network radiation.
- (iii) Equation 2 the math function term is missing please insert.
- (iv) p7, L14, L20 need to be consistent in referring to figures either with a capital Fig or

lower case fig.

(v) p7, p.30 need to add equation number.

(vi) Figure 6 right hand side scale label.

(vii) p7,L27 full stop at "seen".

(viii) p8, L29 I can clearly see $l=+1$ mode but the $l=-1$ mode is not strongly excited and also doesn't appear to exist for all frequencies. Explain why?

(ix) The abstract states that you wish to excite and use the LG $l=-1$ mode however the $l=+1$ seems to dominate the mode spectrum in Figure 6 and forms the basis for the subsequent results in Fig. 7. Please explain?

(ix) p10,L17 The authors should comment or compare the demonstrated spectral efficiencies per mode in context with channel capacity from an equivalent patch antenna array radio link.

===PREPARING YOUR MANUSCRIPT===

===PREPARING YOUR REVISION IN SCHOLARONE===

-- Ensure that your data access statement meets the requirements at <https://royalsociety.org/journals/authors/author-guidelines/#data>. You should ensure that you cite the dataset in your reference list. If you have deposited data etc in the Dryad repository, please only include the 'For publication' link at this stage. You should remove the 'For review' link.

-- If you have uploaded ESM files, please ensure you follow the guidance at <https://royalsociety.org/journals/authors/author-guidelines/#supplementary-material> to include a suitable title and informative caption. An example of appropriate titling and captioning may be found at https://figshare.com/articles/Table_S2_from_Is_there_a_trade-off_between_peak_performance_and_performance_breadth_across_temperatures_for_aerobic_sc_ope_in_teleost_fishes_/3843624.

Author's Response to Decision Letter for (RSOS-201711.R0)

See Appendix A.

Decision letter (RSOS-201711.R1)

Dear Dr Allen,

It is a pleasure to accept your manuscript entitled "Experimental Evaluation of Spectral Efficiency from a Circular Array Antenna producing a Laguerre-Gauss Mode" in its current form for publication in Royal Society Open Science.

on behalf of Dr Chong Li (Associate Editor) and R. Kerry Rowe (Subject Editor)
openscience@royalsociety.org

Appendix A

Thank you for the time the editors and reviewers have spent giving our paper your consideration. We have updated our manuscript according to the requests as outlined below. We hope you now find it fully find it suitable for publication.

With kind regards

Ben Allen, Tim Drysdale, Chris Stevens,

+++++

Associate Editor Comments to Author (Dr Chong Li):

Comments to the Author:

Dear Authors,

The paper demonstrates experimental evaluation of spectral efficiency for a circular antenna array with LG mode of radiation. This is interesting work and could potentially benefit new developments of antennas of this kind and therefore contribute to the development of wireless communications. This work should be published. However the reviewers have some comments especially Reviewer 2's questions ii, viii-ix that should be addressed carefully by the authors.

Associate Editor

Dr. Chong Li

Thank you for your positive comments. All reviews comments have been addressed, including those relating to reviewer 2.

Reviewer: 1

Comments to the Author(s)

In this paper authors presented an interesting concept of 4D volumetric electromagnetic field measurements. Results are validated by measurements. Some minor comments are:

- Add some quantitative results in the abstract.

Done. Please see updated manuscript.

- A comparison with state of the art should be added.

Please see references 9-12 in the introduction which describes prior relevant art. The 'best' achievement so far has been by NTT where the team have achieved 100Gb/s using 10 modes [ref 12].

-Dimensions should be added in the figure 4 for each element.

Done. Please see updated manuscript.

-reference for equ 2 is missing

Done. Please see updated manuscript.

-legends in fig 5 are not readable

Apologies, we have not made any amendments as they appear readable in our version. I am happy to work with the production team on any such changes they request.

Reviewer: 2

Comments to the Author(s)

The paper presents an experimental communication radio link spectral efficiency performance for a circular patch antenna array radiating the Laguerre-Gauss mode ($l=1$) over the 9-10 GHz. The results make a demonstrative and significant contribution to the application and performance of BPSK, QPSK and QAM modulation schemes using OAM (orbital angular momentum).

However, there are a number of minor typographical and presentation issues that need attention and clarification as follows;

(i) p3, L55, need a full stop after "patterns".

Done. Please see updated manuscript.

(ii) What is the impact on the radiation/polarisation pattern of the LG ($l=-1$) mode due to the corporate feed network radiation.

Done. Please see updated manuscript.

(iii) Equation 2 the math function term is missing please insert.

Done. Please see updated manuscript.

(iv) p7, L14, L20 need to be consistent in referring to figures either with a capital Fig or lower case fig.

Done. Please see updated manuscript.

(v) p7, p.30 need to add equation number.

Done. Please see updated manuscript.

(vi) Figure 6 right hand side scale label.

Done. Please see updated manuscript.

(vii) p7,L27 full stop at "seen".

Done. Please see updated manuscript.

(viii) p8, L29 I can clearly see $l=+1$ mode but the $l=-1$ mode is not strongly excited and also doesn't appear to exist for all frequencies. Explain why?

Done. Please see relevant place in updated manuscript.

(ix) The abstract states that you wish to excite and use the LG $l=-1$ mode however the $l=+1$ seems to dominate the mode spectrum in Figure 6 and forms the basis for the subsequent results in Fig. 7. Please explain?

Done (typo). Please see updated manuscript.

(ix) p10,L17 The authors should comment or compare the demonstrated spectral

efficiencies per mode in context with channel capacity from an equivalent patch antenna array radio link.

Done. Please see relevant place in updated manuscript.